# The Tumor Proteolytic Landscape: A Challenging Frontier in Cancer Diagnosis and Therapy

**DOI:** 10.3390/ijms22052514

**Published:** 2021-03-03

**Authors:** Matej Vizovisek, Dragana Ristanovic, Stefano Menghini, Michael G. Christiansen, Simone Schuerle

**Affiliations:** Department of Health Sciences and Technology, Institute for Translational Medicine, ETH Zurich, CH-8092 Zurich, Switzerland; vizovisek@imsb.biol.ethz.ch (M.V.); dragana.ristanovic@hest.ethz.ch (D.R.); stefano.menghini@hest.ethz.ch (S.M.); michael.christiansen@hest.ethz.ch (M.G.C.)

**Keywords:** tumor microenvironment, protease activity, protease diagnostic and therapeutic modalities

## Abstract

In recent decades, dysregulation of proteases and atypical proteolysis have become increasingly recognized as important hallmarks of cancer, driving community-wide efforts to explore the proteolytic landscape of oncologic disease. With more than 100 proteases currently associated with different aspects of cancer development and progression, there is a clear impetus to harness their potential in the context of oncology. Advances in the protease field have yielded technologies enabling sensitive protease detection in various settings, paving the way towards diagnostic profiling of disease-related protease activity patterns. Methods including activity-based probes and substrates, antibodies, and various nanosystems that generate reporter signals, i.e., for PET or MRI, after interaction with the target protease have shown potential for clinical translation. Nevertheless, these technologies are costly, not easily multiplexed, and require advanced imaging technologies. While the current clinical applications of protease-responsive technologies in oncologic settings are still limited, emerging technologies and protease sensors are poised to enable comprehensive exploration of the tumor proteolytic landscape as a diagnostic and therapeutic frontier. This review aims to give an overview of the most relevant classes of proteases as indicators for tumor diagnosis, current approaches to detect and monitor their activity in vivo, and associated therapeutic applications.

## 1. Introduction

Dysregulated proteolysis, elevated protease expression, misfiring of protease signaling, or distorted protease-inhibitor equilibrium are frequently associated with developing or ongoing disease [1]. In healthy cells, proteases are instrumental for protein processing, metabolism, coagulation, tissue remodeling, homeostasis, programmed cell death and autophagy, antigen presentation, and immune response, among other physiological functions. Together, proteases represent one of the largest protein families, with a total of around 580 genetically encoded hydrolytic enzymes in humans [2], divided into the five major families of metallo, serine, cysteine, aspartic, and threonine proteases based on their catalytic mechanism [3,4]. While members of the same protease family often display a substantial degree of similarity in terms of structure and sequence homology, each individual protease has its own unique specificity fingerprint, activity patterns, expression profiles and localization [5]. Before the extensive developments in the fields of molecular and cell biology and the advent of the omics era in 1990s, proteases were essentially considered as protein-degrading enzymes instrumental for metabolic processes and maintenance of cell homeostasis [6]. Advances in the fields of molecular biology, chemical biology and proteomics have challenged this view, and proteases are now widely recognized as major players in diseases and considered important drug targets [7,8].

Past research provided solid evidence that proteases are heavily involved in the development and progression of cancer [9] and efforts to integrate them into cancer diagnosis and therapeutic management have grown [10]. In cancer, dysregulated proteases from different protease families are associated with a myriad of stages of development, extensive remodeling of the ECM (Extracellular matrix) [11,12,13], epithelial-to-mesenchymal transition [14], immune system evasion and hijacking [15], resistance to apoptosis signals [16], metastasis development [17,18,19], as well as tumor growth, invasion and metastasis spread [20,21,22]. Furthermore, proteases have roles in signaling pathways like MAPK (Mitogen-activated protein kinase), Akt (Protein kinase B) and TNFβ (Tumor necrosis factor β), among others, thereby exerting a substantial influence on cancer development and progression [23,24]. With the growing knowledge of protease molecular functions and characteristic activity patterns associated with cancer phenotypes, it has become clear that tumor-associated protease activity can be harnessed to develop diagnostic tools and biomarkers for early disease detection [10,25]. Moreover, the activity of disease proteases can be exploited for functional diagnostic imaging [26,27], translate into applications where proteases act as activators of prodrugs [28] or into protease-based drug delivery systems [29] to pave the way towards the next generation of clinical modalities as outlined in Figure 1. 

Here, we review the most important roles of each major catalytic type of protease in the development and progression of cancer. We describe the tools commonly used to detect and monitor their activity in preclinical settings and illustrate how this proteolytic landscape can be integrated into diagnostics and therapeutics. Attention is also given to the emerging synergies and interdisciplinary connections with the nanotechnology and nanomaterial field, where recent developments have shown potential to drive the evolution from bench-to-bedside.

## 2. Proteases in the Tumor Microenvironment

Representative members from each major protease family have been linked to selected cancer hallmarks (Figure 1) and elevated or dysregulated protease activity is mechanistically involved in sustaining cancer cell proliferation, resisting cell death, evading growth suppressor signals, supporting replicative immortality, inducing angiogenesis and neovascularization and metastasis development and invasion [21,30]. Accordingly, representative protease groups and their roles in context of the original six cancer hallmarks proposed by Hanahan and Weinberg are summarized in Table 1. Proteolytic cleavage is a relatively simple irreversible posttranslational modification that generates proteolytic fragments from proteins, thereby changing their structure and function. Understanding the complex contribution of proteases to cancer and leveraging this knowledge for clinical purposes is nevertheless a daunting task for several reasons. First, proteases from different families with different proteolytic activities are present in the tumor microenvironment. These can originate either from cancer cells or from other cells that are present in the tumor microenvironment (macrophages, neutrophils, stroma cells etc.) [31]. Second, the concentration of proteases can span several orders of magnitude, from those present in trace amounts to others with high expression levels or locally elevated concentration [32]. Third, proteases show different localization and activity patterns and can be found in multi-protein complexes or in complexes with their inhibitors that fine-tune their activity [9]. Finally, there is a substantial level of crosstalk between proteases from different families in the tumor microenvironment [17,19]. Together, all these factors contribute to an intricate and interconnected tumor proteolytic landscape. In the next sections, we summarize the current knowledge on the role of the five major classes of proteases in the context of the hallmarks of cancer.

### 2.1. Metalloproteases

Metalloproteases are a family of Zn^2+^ binding protease homologues with numerous cancer-related functions and represent the largest protease group in humans. They exhibit diverse roles, functions, and patterns of localization [33,34], and are instrumental for pericellular proteolysis with direct effects on ECM structure, function, and signaling [56]. The most important metalloproteases that are highly active in the tumor microenvironment are the matrix metalloproteases (MMPs), a disintegrin and metalloproteases (ADAMs), and a disintegrin and metalloproteinase with thrombospondin motifs (ADAMTSs). In a healthy state, metalloprotease activity is generally kept under tight regulation by TIMP (Tissue inhibitor of metalloproteinases, TIMP-1, -2, -3 and -4), which fine-tune the rate and extent of target protein proteolysis [75]. Any disruption of this equilibrium can unleash the proteases’ degradative power, with MMPs having particularly detrimental effects in the context of cancer [76]. 

Generally, MMPs target a broad range of ECM proteins, contributing to cancer development, progression, invasive growth and spread of cancer cells, and their elevated activity has so far been detected in almost all types of cancer [40]. Traditionally, MMPs are divided into groups of collagenases, gelatinases, stromelysins and matrilysins according to their effect on the ECM proteins [34,57,77]. MMP-1, -8 and -13 are strong collagenases that cleave triple helix collagens, while MMP-1 and -8 also cleave gelatin [78,79]. MMP-3, -10 and -11 belong to stromelysins, cleaving different ECM proteins like aggrecan and fibronectin, but do not cleave collagen. MMP-7 and MMP-26 are matrilysins that cleave gelatins, collagen and fibronectin, whereas MMP-2 and MMP-9 are gelatinases [34,57,77]. The proteolytic action of MMPs on these scaffolding proteins changes composition, structure and function of ECM [80]. Another important group of MMP substrates are cell adhesion molecules like syndecans or E-cadherin [81,82] and non-ECM substrates [83], and products of these cleavages are linked with disease development and progression. Interestingly, several MMPs (as well as serine proteases) can contribute to activation of other MMPs, amplifying their proteolytic activity in the ECM [84]. MMPs can also play a role in cell signaling and have pleotropic roles in cancer [85], such as the ability of MMP-7 to activate different kinase pathways [86]. In addition to soluble MMPs, there are six membrane-bound MMPs, affixed to the cell surface by a GPI-anchor. These include MMP-14, -15, -16, -17, -24 and -25, which have important roles in remodeling the ECM, processing growth factors, and shedding cell receptors, which drive cancer development and progression [61]. MMPs are also involved in the epithelial-to-mesenchymal transition, one of the most widely-recognized cancer hallmarks [62,63]. Furthermore, correlations discovered between MMPs and poor patient prognosis suggest that MMP levels could be used as a diagnostic or prognostic indicator [87,88,89,90,91]. 

ADAMs are transmembrane proteases with established roles in cell proliferation, adhesion and migration, and substantially contribute to the complexity of proteolysis in the tumor [35,36]. Whereas MMPs essentially target most of the structural components of the ECM, ADAMs mostly target the extracellular domains of transmembrane proteins (both type I and type II) and thus contribute substantially to the cleavage of cell adhesion molecules, shedding of cell surface receptors and maturation of cytokines and chemokines [36,47,48]. Currently, ADAM-10 and ADAM-17 present the strongest bodies of evidence for involvement in cancer. ADAM-10 is able to process and activate the EGFR (Epidermal growth factor receptor) ligands [92], cleave E-cadherin with implications in signaling [93], and cleave the CD44 adhesion molecule from the cell surface [94]. ADAM-10 expression was also shown to correlate with invasive growth of different cancers [95], establishing the protease as a potential therapeutic target. ADAM-17 is important for the release of the soluble TNFα (Tumor necrosis factor α) [96] and activation of IL-6/ERK signaling [97]. It can release the EGF receptor [98] and shed adhesion molecules such as CD44 [99]. While inhibition of ADAM-17 can reduce invasiveness of breast cancer [100], its activation as a consequence of chemotherapeutic drugs can contribute to resistance [101]. Different ADAMs, including ADAM-10 and ADAM-17, are also important for proteolytic processing of ligands and modulation of Notch signaling. This molecular mechanism requires ADAM-mediated cleavage events for its proper functioning and altered processing of Notch receptors and ligands has been linked to proliferation and differentiation as well as cancer cell death [102,103,104].

Another family of metalloproteases with roles in cancer are ADAMTSs, largely responsible for degrading structural ECM proteins [105]. They are generally divided into the group of hyalectanases, including ADAMTS-1, -4, -5, -8, -9, -15, and -20, which mostly target various (hyalectan) proteoglycans, ADAMTS-2, -3 and -14 which process the N-terminal propeptides of collagens, and the remaining group of ADAMTS-6, -7, -10, -12, -13 and -16, which have more specific functions [106,107,108]. Accumulating evidence suggests several ADAMTSs are involved in cancer development and progression and can promote tumor development [49,50,51], but their roles in proteoglycan proteolysis also have tumor suppressor functions [109] and their prognostic value was evaluated in the context of different cancers [110,111,112]. Since the involvement of different members of the metalloproteinase families has been reported in nearly all known forms of cancer, they are considered one of the most important target protease groups in cancer research.

### 2.2. Serine Proteases

Serine proteases comprise the second largest family of proteolytic enzymes, and execute functions ranging from metabolism and blood coagulation to homeostasis and immune response. They are mostly secreted, activated by limited proteolytic processing, and tightly regulated by their endogenous inhibitors, SERPINS [113]. Disturbance of this equilibrium can be a factor contributing to cancer development, progression, metastasis and invasion [114]. Trypsin and trypsin-like proteases (e.g., thrombin or tissue factor) are perhaps the most comprehensively characterized serine proteases. While assuming essential roles in metabolism, the coagulation cascade and blood pressure regulation, they have also been linked to cancer where hemostatic abnormalities are often observed [115]. The tumor-induced activation of the coagulation cascade can result in elevated levels of active coagulation proteases [116] with an impact on tumor growth and angiogenesis [117,118]. It is therefore unsurprising that elevated pro-coagulant activity was observed in cancer patient samples [119]. Currently, thrombin is considered a potential therapeutic target in lung cancer that could contribute to cancer progression [120]. In addition, tissue factor has been associated with aggressive behavior of malignancies and a substantial body of evidence suggests that coagulation proteases participate in activation of PARs (protease-activated receptors) to propagate tumor growth and invasiveness [121,122]. Notably, there is appreciable crosstalk between the coagulation proteases and the immune system and the activation of one can boost the activation of the other [123].

Immune response, or the lack thereof, is central in allowing cancer to develop, and several serine proteases are involved in immune response. Granzyme B is a serine protease from the natural killer cells and cytotoxic T lymphocytes [41,42]. It is important for the removal of tumor cells by the host immune system [124] and for different apoptosis mechanisms that contribute to the removal of cancer cells [125]. Generally, elimination of tumor cells is strongly dependent on granzyme B and this process can be enhanced by activating p53 to support tumor cell lysis [126]. Nevertheless, granzyme B can also contribute to cancer cell invasion by degrading ECM components [127]. Neutrophil elastase (ELANE), secreted into the tumor microenvironment primarily by immune cells like neutrophils and macrophages, has been shown to be upregulated in cancer [128]. ELANE can contribute to the remodeling of ECM [129], activation of toll-like receptors (e.g., TLR4) [130] and release of growth factors like TGFα (Transforming growth factor α), PDGF (Platelet-derived growth factor) and VEGF (Vascular endothelial growth factor) [131]. Another neutrophil protease involved in cancer is cathepsin G. It has been shown to be a mediator of MMP-9 activation and promotes TGFβ (Transforming growth factor β) signaling that is important for formation of bone lesions, cancer-induced osteolysis and metastasis [132,133]. Cathepsin G is also involved in perturbing E-cadherin-dependent cell adhesion, linking it to cancer cell migration and invasiveness [68,69]. Furin, a member of the subtilisin-like family has also been linked to cancer [134]. Its activity has been detected in various cancers and can promote the invasive phenotype, making it a potential therapeutic target [135].

Kallikreins, the largest subgroup of serine proteases, represent yet another class of serine proteases that appear to be implicated in cancer. Accounting for a total of 15 proteins in humans, KLK1–KLK15 are generally expressed within endocrine glands and organs and ultimately secreted in the extracellular space [38,136]. They are emerging as potential diagnostic targets in strategies for monitoring chemotherapeutic response [39] and there is evidence for their value as clinical biomarkers [137,138]. While the spectrum of functions of kallikreins is diverse, they appear to be essential in the development of prostate and skin cancers [52,139]. KLK4, 5, 6 and 7 are involved in TGFβ signaling as demonstrated by degradome analysis [60]. Kallikreins have important links between ECM proteolysis and infiltration of immune cells [53] and almost all kallikreins can cleave structural components of the ECM [140]. They are also involved in PAR and EGF receptor signaling [141]. PSA (Prostate-specific antigen), a protease very similar to the members of the kallikrein family, is typically expressed in prostate cells and routinely measured in clinics as a biomarker for diagnosis of prostate cancer [142]. PSA is key in ECM remodeling and signaling pathways associated with prostate cancer progression, metastasis and angiogenesis [64]. PSA is also capable of activating uPA (urokinase-type plasminogen activator) [143]. uPA is a trypsin-like protease that catalyzes the activation of uPAR (urokinase-type plasminogen activator receptor) and cleaves several components of ECM including fibronectin [144]. Currently, different components of the uPA/uPAR system are under consideration as prospective cancer biomarkers [145].

Moving away from trypsin-like serine proteases, DPPIV (Dipeptidyl peptidase IV), FAP (Fibroblast activation protein) and PEP (Prolyl endopeptidase) have emerged as potentially important players in cancer [70]. DPPIV seems to be closely linked to malignancies, though it has been suggested to have both cancer suppressing and promoting roles, indicating that further investigation is needed [146]. Notably, DPPIV serum levels have shown prognostic value in patients with colorectal [147] and gastric [148] cancer. FAP, which has strong gelatinase activity implicated in extensive remodeling of ECM, has been linked to elevated invasiveness and tumor progression [149,150]. Elevated PEP activity levels have also been found in various cancers [151] and tend to correlate with poor patient prognosis, as demonstrated in colorectal cancer [152].

Another group of serine proteases with possible links to cancer are the High-temperature requirement factors, HtrA1, 2, 3 and 4 [153]. From this group HtrA2 has emerged as the most promising target because of its role in apoptosis regulation [43]. There are also several membrane-anchored serine proteases, including matriptase and Hepsin [154]. Matriptase is considered a potential diagnostic marker [155] and has an important role in the serine protease—growth factor signaling axis [156]. Hepsin, a transmembrane protease, has received attention because of its overexpression in prostate [157] and gastric cancer [158], which correlated with poor patient prognosis.

Finally, there is the group of rhomboids [159]. These intramembrane proteases usually cleave transmembrane proteins to release their domains from the cell membrane. The most important of the rhomboids are RHBDL2 (Rhomboid-related protein 2), which can trigger the activation of the EGF receptor [160] and has been found to be highly expressed in breast cancer [161] and RHBDL4 (Rhomboid-related protein 4), which triggers non-canonical secretion of TGFα [162]. Rhomboids have been linked to angiogenesis [163], resistance to cell death, and proliferation of cancer cells [164]. Research interest in better understanding their functions has been increasing.

### 2.3. Cysteine Proteases

The cysteine proteases that appear to be most directly involved in cancer are cathepsins, caspases and calpains. Cysteine cathepsins are a family of 11 proteases with a papain-like fold and a catalytic Cys-His amino acid pair in their active site. While cathepsins B, C, H and X are exopeptidases, cathepsins F, K, L, O, S, V and W are potent endopeptidases [65]. Cysteine cathepsins are generally regarded as lysosomal proteases and their extra-lysosomal and extracellular activity has previously been linked to various aspects of cancer development and progression [37]. Activity of cysteine cathepsins is regulated by pH, localization, proteolytic degradation and by their protein inhibitors (cystatins and stefins). A substantial body of evidence links cathepsins B, C, H, K, L, S and X to various aspects of cancer, making them potentially useful biomarkers [58]. Their oncologic roles include remodeling the ECM via cleavage of structural proteins, altering the signaling pathways that govern cell growth, proliferation and cell death or helping to fuel the protease pool that drives chronic inflammation [54]. Cysteine cathepsins are mostly active in acidic conditions like the tumor microenvironment and can cleave many different proteins, ranging from components of the extracellular matrix like collagens, elastins, laminins, glycosaminoglycans and proteoglycans to various cell adhesion molecules [59]. These belong to groups of junction adhesion molecules (JAM) [165] and cell surface receptors (like the EGF receptor, plexins and neuropilins) [166]. Importantly, the extracellular proteolysis by cathepsins in the tumor microenvironment has been linked to cancer spread, altered cell adhesion, neovascularization, metastasis, and invasive growth [18,167,168] and evidence supports cathepsin secretion as one of the driving forces behind cancer progression [55]. Findings from mouse cancer models suggest that general degradation dominates over specific proteolysis in tumors [169]. Nevertheless, cathepsins can also cleave specific targets like chemokines and cytokines that are linked to inflammation [170]. While the elevated activity of cysteine cathepsins in cancer, especially in the extracellular space, is now widely recognized as a characteristic sign of the disease, their redundancy [171] and largely overlapping specificity features [172,173] make their direct translation into diagnostic and therapeutic modalities a challenge.

Caspases are a group of endopeptidases that cleave proteins specifically after aspartate. Generally, caspases are divided in groups of apoptotic and inflammatory caspases (caspase-1, -4, -5 and -12 in humans). Apoptotic caspases are further divided in initiation (caspase-2, -8, -9 and -10) and executioner caspases (caspase-3, -6 and -7) and are essential for facilitating programmed cell death [174]. An important hallmark of oncologic transformation is also the cancer cells’ resistance to programmed cell death, for instance by insensitivity towards apoptosis triggers [44,175]. Past studies have established strong links between levels of caspase expression and the sensitivity of cancer cells towards apoptosis. Moreover, caspase deregulation generally contributes to resistance towards therapeutic intervention [45]. Nevertheless, for several caspases, their exact cancer-related roles depend heavily on oncologic context and often involve functions that are not directly related to apoptosis [176,177,178]. This creates substantial difficulties for caspases as potential biomarkers, as demonstrated by caspase-3. While some studies found that the expression levels of caspase-3 correlated with poor prognosis for the disease outcome [179], others reported that the levels of caspase-3 in cancer samples were extremely low or even non-detectable [180], suggesting that further activity studies are needed to fully understand the role of caspases and to evaluate them as prospective disease biomarkers. Inflammatory caspases have also been linked to cancer, where especially caspase-1 and inflammasome activation can lead to inflammation-driven cancer development and progression [181,182]. Another notable cysteine protease that shares structural features with caspases is legumain. It is an asparaginyl endopeptidase highly expressed in breast, prostate, gastric and ovarian cancer [71]. Beside cancer cells, it is highly expressed by tumor-associated macrophages [183] and generally correlates with poor disease prognosis [184,185].

The third group of cysteine proteases are the 14 calpains, calcium-activated proteases with important roles in remodeling of ECM, regulation of apoptosis, and different cell signaling pathways [67]. They are involved in several aspects of tumor cell invasion, metastasis formation, and cancer spread [73]. Interestingly, there is evidence for calpains roles in cancer cell death and survival with important links to signaling pathways involved in cancer development and progression [74]. They are also implicated in activation of apoptosis and could be potentially interesting targets for chemotherapy-induced apoptosis of cancer cells [186,187]. Calpains were linked to poor outcomes in breast [188], pancreatic [189] and ovarian cancer [190], among others, and they are currently considered an important drug discovery frontier [66]. While the cysteine protease pool within the tumor microenvironment generates a complex proteolytic landscape, it is likely that even closely related disease phenotypes will have a characteristic protease activity fingerprint that could support precise disease diagnosis and staging.

### 2.4. Aspartic Proteases

In humans, the most important aspartic proteases with links to cancer are renin, cathepsins D and E, pepsin C, and napsin A. The family is characterized by two catalytic aspartic acid residues in the active site. Renin is the essential player in the RAS (Renin-angiotensin system), which regulates the blood pressure [191]. While elevated activity of RAS can result in hypertension, there is also evidence for involvement of the system in cancer development and progression by influencing tissue remodeling, inflammation, cancer cell proliferation and apoptosis [192]. Perturbations of the RAS system are linked to pathways that are deregulated in the pre-cancer stage and contribute to malignant transformation [193]. Furthermore, the RAS can impact immunosuppression in tumors [194] and activation or inhibition of the system has been associated with different cancers [195]. These findings present an opportunity to leverage knowledge of RAS to potentially improve cancer therapies [196]. 

Moving to aspartate cathepsins, cathepsin D is a protease residing in lysosomes under normal physiological conditions and is instrumental for protein degradation. It has been linked to multiple cancer related processes with experimental results showing its role in tumor progression, angiogenesis and apoptosis [46]. Increased expression and secretion of cathepsin D was observed in several cancers, including malignant melanoma, prostate, ovarian and breast cancer [197]. Elevated serum levels of cathepsin D were reported in breast cancer patients [198] and cathepsin D detected in tissues could have diagnostic value for ovarian cancer [199]. While serum levels of cathepsin D have been previously investigated as a biomarker with inconclusive results, its activity patterns could be potentially used to develop early diagnostics. Another promising candidate is cathepsin E, an intracellular protease expressed mostly by immune cells and in the gastrointestinal system, important for antigen processing/presentation, apoptosis, cytokine turnover and adipose tissue regulation [72,200]. Even as early as two decades ago, high levels of cathepsin E in pancreatic juice were suggested to be indicative of adenocarcinoma of the pancreas [201]. High levels of cathepsin E are characteristic for lesions found in pancreatic adenocarcinoma and could be a valuable biomarker for early detection of pancreatic cancer [202], which is still very challenging to diagnose at an early stage. 

Pepsin C belongs to the group of common digestive enzymes of the gastrointestinal system, mostly present in the stomach from gastric mucosa cells. Pepsin C is essential for normal digestive processes, and significant changes in its expression levels were detected in breast, prostate and ovarian cancer [203,204]. Nevertheless, its diagnostic applications are currently very limited. Another aspartic protease similar to pepsin is napsin A, which is important for processing surfactant B in lungs [205] and is an established biomarker used in diagnosis of lung adenocarcinoma [206]. Despite the fact that aspartic proteases received less attention than other protease families in the context of cancer, new tools for monitoring their activity could be beneficial for their stratification as clinical biomarkers.

### 2.5. Threonine Proteases

The most important threonine proteases linked to several aspects of cancer development and progression are the proteasomes. These are multi-protease complexes composed of several subunits that efficiently and non-selectively degrade most of the cellular proteins marked for degradation by the ubiquitin-conjugation system [207]. Proteasomes have three different types of catalytic subunits with characteristic proteolytic activities. The b1 subunit has a caspase-like activity, the b2 subunit has a trypsin-like activity and the b5 subunit has a chymotrypsin-like activity as investigated with combinatorial peptide libraries [208]. While proteasomes are mostly responsible for nonspecific protein degradation, there is substantial evidence for their involvement in cancer. Proteasome inhibition is considered an important therapeutic strategy because of its central role in regulating cell homeostasis. Generally, this inhibition leads to cancer cell apoptosis by accumulation of pro-apoptotic proteins and disruption of the NF-κβ (Nuclear factor κΒ) pathway. There are currently several proteasome inhibitors at various stages of testing in clinical trials [209]. Inhibitors like bortezomib have been approved by the FDA for treatment of multiple myeloma [210]. With ongoing efforts to develop new compounds for targeting the proteasome in cancer, there is substantial interest in gaining a better understanding of how multiple proteasome activities could be exploited in clinical settings.

## 3. Monitoring Protease Activity in Cancer

The proteolytic landscape of cancer consists of proteases representing different families, displaying different activity and patterns of localization, exhibiting varying degrees of substrate specificity, and acting upon considerably overlapping pools of natural substrates. All of this leads to nuanced pathological protease fingerprints, often characteristic for a specific disease phenotype. While this inherent complexity poses a formidable challenge for detection of proteases in the context of cancer, it also offers opportunities to harness and leverage specific pathologic protease activity as a basis for sensitive new tools for precision medicine [10] as outlined in Figure 2, left.

### 3.1. Activity- and Substrate-Based Probes

Elevated protease activity can be successfully translated into activity-based probes and protease substrate reporters [211]. Both technologies aim to reveal the location and the amount of protease activity, accomplishing this via different strategies. Activity-based probes typically bear a warhead that specifically reacts with the active site of the protease and a reporter group that generates a signal upon interaction with the protease. This chemical biology approach has found broad application in visualization of cancer cells, tissues, and protease activity patterns. In 2006, Sieber et al. applied cocktails of active-site probes in combination with mass spectrometry for a comprehensive profiling of the metalloprotease family [212]. Fluorescent activity-based probes have been reported for all major protease families to date. Examples include metalloproteases [213,214,215], serine proteases [216] (e.g., neutrophil proteases [217] or inflammation-related serine proteases [218]), cysteine proteases (e.g., cathepsins [219,220], caspases [221,222] or legumain [223,224]), aspartic proteases [225], and recently also for multi-domain proteasomes [226,227]. Activity based probes have substantially improved our understanding of the role of proteases in cancer and the tumor microenvironment, as demonstrated by a cathepsin S activity-based probe applied to image tumor-associated macrophages [228]. Dual color activity-based probes were also developed to monitor the localization of cathepsin S activity to elucidate its cancer functions [229]. Activity-based probes can exactly pinpoint the protease location, but they have the disadvantage of inactivating the protease upon binding, which can not only cause unwanted perturbations in the experimental system, but also precludes the possibility of signal amplification. Another major challenge is to design selective probes that have good specificity towards the target enzyme without interference from other closely-related proteases [230].

Substrate probes exploit protease cleavage on the target site, usually to generate a fluorescent signal [231,232,233]. The majority of substrate probes are designed either as quenched substrates that fluoresce more intensely after cleavage or as FRET (Förster resonance energy transfer) probes in which cleavage causes a shift in the emission spectra that can be precisely quantified as a measure for protease activity. Both detection concepts have been successfully applied to cancer-related settings [234]. Selective substrates can be designed using substrate libraries like PS-SCL (positional scanning substrate combinatorial library) [235], CoSeSuL (counter selection substrate library) [236] or HyCoSuL (hybrid combinatorial substrate library) [237] and these strategies were successfully applied to caspases [222], cathepsins [238,239], neutrophil proteases [217] and kallikreins [240]. Recently, a HyCoSuL-based assay was used for screening protease activity in biopsies [241].

An alternative to these combinatorial approaches to design protease probes is to convert a pharmacologically optimized inhibitor to a substrate by replacing the warhead with a protease-cleavable peptide. This has been successfully demonstrated for luminescent probes for caspase-1 [242]. In general, substrate probes have the advantage that they do not inactivate the target protease and thus do not immediately influence on-site protease activity. The protease can thus cleave further substrate molecules, leading to signal amplification, an effect that is counteracted by diffusion away from the target site. One approach to address this problem has been to improve the reporter on-site retention after protease cleavage [243].

### 3.2. Integrating In Vivo Imaging Modalities

Synergies with the field of nanomaterials have shown great potential for the development of various protease-sensitive nanomaterials suitable for optical imaging modalities [244]. For example, QDs (quantum dots) consisting of ZnS or CdSe can be used as fluorescent reporters and these nanomaterials have a broad spectrum of application for in vivo imaging studies, including imaging of protease activity [245]. Like fluorescent molecules, QDs exhibit FRET effects, and several different FRET-based QD systems have been prepared and utilized as protease sensors for multiplexed tracking of protease activity, such as monitoring trypsin and chymotrypsin [246]. QDs exploit the FRET effect between the quantum dot and a suitable fluorescent dye bound by a protease-cleavable peptide, resulting in fluorescence emission spectrum change upon protease cleavage. This concept was used to design protease reporters for MMPs [247,248], caspase-1, collagenase, chymotrypsin and thrombin [249], uPAR [250], caspase-3 [251] and kallikrein [252]. Recently, the concept was also extended for monitoring MMP-2/-9 activity in the tumor microenvironment [253]. While these concepts can be employed for sensitive monitoring of clinically relevant proteases, they are very difficult to translate in modalities that would be applicable to clinical settings, mostly due to high costs and toxicity of materials for QD preparation. Fortunately, there are also non-toxic materials like silicon that can be used to make QDs [254], but even comparatively biocompatible QDs cannot not overcome the limitations of optical imaging.

Various classes of probes described in the previous sections have substantially advanced understanding of cancer mechanisms and can be applied in a broad range of contexts, from labeling cell lysates, intact cells to ex vivo, and in vivo imaging of small animals [255]. Nevertheless, a broad translation of fluorescent probes into clinical modalities has not yet occurred, despite the fact that NIR (near-infrared) fluorescent protease reporters have shown great potential for more than a decade [256]. Image-guided surgery with NIR fluorescent probes is an emerging application that seems especially well-poised for clinical translation. For example, a cathepsin-sensitive poly(L-glutamic acid)-based quenched fluorescent probe, Prosense^®^ 680, which is commercially available from Perkin Elmer, was successfully used for detection of tumor margins in image guided surgery [257]. In a recent report, another cathepsin-sensitive quenched fluorescence activity-based probe designed for intravenous application was used to visualize surgical margins of the tumor and thus increase the probability of its complete removal [258]. Besides image-guided surgery, the NIR reporters have potential to translate into diagnostic assays as demonstrated by FAP and PREP [259]. A NIR FRET-based probe LUM015 sensitive for cysteine cathepsins has been tested in a mouse-human phase I co-clinical trial, offering the first promising human data for this technique [260].

Unfortunately, the use of fluorescent probes for deep tissue imaging is inherently precluded by the limitations of fluorescence as a readout. Nevertheless, the concepts developed for selective protease detection with fluorescent reporters are transferrable to other detection modalities [261]. Here, PET (positron emission tomography), SPECT (single-photon emission computed tomography), CT (computed tomography) and MRI (magnetic resonance imaging) have proven to be valuable technologies for noninvasively tracing protease activity. In PET and SPECT, a typical protease tracer consists of a radioactive isotope like ^11^C and ^18^F for PET or ^123^I and ^131^I for SPECT and CT. These tracers require not only excellent selectivity towards the target protease, but also good clearance properties to avoid background interference [262]. The half-lives of isotopes used in PET or SPECT tracers are usually short, posing a unique barrier to commonplace application of these modalities for protease imaging. Nevertheless, the technology was successfully demonstrated for imaging metalloproteases [263,264,265], cysteine cathepsins [266,267,268] and caspases [269,270] in preclinical settings. In CT imaging modalities, X-rays are used to create cross-sectional images with high resolution and this technique is widely used for noninvasive clinical imaging [271]. Contrast agents are an integral part of any CT imaging, but development of protease-sensitive molecules for CT applications has proven challenging. Recently, cancer imaging studies for CT imaging of multiple cysteine cathepsins were performed using a protease-targeted iodinated probe [272] or activity probes based on gold nanoparticles [273].

Another technology applicable to in vivo imaging of proteases is MRI, which detects certain nuclei or protons and is sensitive to interactions with their surrounding molecular context. The influence of molecular context on relaxation times offers a basis for proteolytic activity to be coupled to detectable changes in contrast, producing an informative readout. Paramagnetic Gd^3+^ compounds are conventionally used as contrast agents, and MRI probes have been developed to visualize protease activity. For example, a Gd-DOTA (Tetraazacyclododecane tetraacetic acid) probe was conjugated with a peptide cleavable by MMP-2, leading to solubility changes and allowing MMP-2 activity to be imaged in tumors [274]. Also, a Gd-based caspase drug was used to monitor caspase activity after drug-induced apoptosis [275]. Examples also extend into MRI modalities employing specialized contrast agents, including ^19^F MRI, in which Gd acts as a quencher [276], and Overhauser-enhanced MRI, which essentially combines MRI with electron paramagnetic resonance to reveal the cleavage of nitroxide-labeled macromolecules [277]. MRI protease-sensitive imaging is clearly a subject rich with possibilities and potential for further expansion, though it remains to be seen which modalities and contrast agents will ultimately be most translatable. Synergies with nanotechnology are one promising area, best typified by a cleavage-responsive nanosensor based on Gd complexes bound to magnetic nanoparticles, revealing MMP-2 activity in a rodent tumor model [278]. Protease-cleavable PLG-based MRI probes (poly-L-glutamate) have also been used in a rodent tumor model, mapping cysteine cathepsin expression [279]. A smart, self-assembling contrast agent developed for furin was also used to detect the protease in cell cancer models [280].

Additional advantages have been realized by combining multiple detection modalities simultaneously, resulting in probes that achieve better resolution and more sensitive detection of the target proteases. This is perhaps best exemplified by dual modality probes designed for several members of the MMP family. For example, NIR/PET probes sensitive for MMP-2, -9 and -13 were used for detection of tumors in a mouse cancer model [281], a FRET/SPECT probe designed for MMP-2 and was used for in vivo imaging of metastatic lymph nodes [282] and a FRET/MRI probe was developed for MMP-2 and bimodal imaging of gastric tumors [283]. There are several other examples of probes with protease-cleavable peptides enabling fluorescent and MRI imaging of proteases [284]. While there is still a dearth of technologies that could be applied for routine, cost-efficient imaging of protease activity in deep tissues, there have also been recent developments in the field of biosensors for acoustic enzyme detection [285].

### 3.3. New Developments and Trends

The methodologies described above can produce high-resolution data of spatiotemporally-resolved protease activity, but the technologies are technically demanding and cost-intensive, often precluding their routine use. Clearly, the continued development of alternative protease detection technologies is needed. Antibodies, one example of an alternative to activity-based probes and other contrast agents, can achieve excellent specificity towards the target protease or even the components of an activity-based probe used for labeling the target protease [286]. Besides antibodies, DARPins (Designed Ankyrin Repeat Proteins) are a useful alternative because of their superior stability and equal or better affinity towards the target [287]. Recently, a highly selective fluorescently-labeled DARPin for cathepsin B was used for in vivo imaging of the protease in cell and animal models of breast cancer [288]. DARPins could include other reporters and thus expand the spectrum of imaging applications.

A concept that has recently shown great potential for translation in clinical settings and is building on the disease-related activity of proteases are the synthetic biomarkers. While classic activity-based tools usually generate a direct readout for one target protease or a very narrow group of closely-related proteases, synthetic biomarkers have the capacity to be applied in multiplexed setups [289]. These nanosensors employ protease-cleavable tagged peptides conjugated to nanoparticles that are cleaved by different disease proteases at the target site (e.g., in tumors), liberating reporters that can be detected in urine by mass spectrometry. Kwong and co-workers demonstrated that detection of urinary biomarkers can outperform the standard CEA (carcinoembryonic antigen) detection in plasma for early diagnosis of colorectal cancer [289]. Also, magnetically actuated protease sensors (MAPS) have been developed. In this assay, magnetic nanoparticles and peptide substrates with fluorescent reporters and biotin affinity tags were co-encapsulated in thermosensitive liposomes, enabling the targeted application of an alternating magnetic field to initiate selective cargo release. Protease activity was then measured by cleaved reporter peptides excreted in the urine, reflecting MMP tumor profiles in colorectal cancer [290]. Knowledge of disease-related protease activity can be integrated into the design of nanosensor libraries that measure protease activity and these ABNs (activity-based nanosensors) have great potential to reveal disease-associated protease activity, as demonstrated by detection of MMP-9, a protease commonly upregulated in human cancers [291]. The concept of synthetic biomarkers is gaining momentum in cancer diagnostic applications and MMP-responsive nanosensors based on AuNC (Ultra small gold nanoclusters) have shown promising results in colorectal mouse cancer models. The AuNC can be renally excreted after cleavage and enable catalytically amplified readouts [292]. Importantly, similar protease nanosensors designed in a bottom-up approach for prostate cancer detection could outperform the PSA prostate cancer marker [293] and a multiplexed substrate panel for lung cancer proteases has demonstrated excellent specificity and sensitivity for disease detection [294]. These encouraging results suggest that the development of future protease activity sensors will be based on synergies with the fields of nanotechnology and nanomedicine. With the development of synthetic biomarkers, it is also essential to develop frameworks to analyze this type of data to extract the maximal amount of meaningful information and aid the application of such diagnostic platforms [295].

Finally, promising results are coming from synergies with microfluidics to enable rapid, multiplexed detection of activity patterns of various proteases with minimum sample requirements. Chen et al. [296] developed a protease assay that uses a picoliter-scale droplet microfluidic platform, utilizing a mixture of inhibitors and FRET substrates and PrAMA (Proteolytic Activity Matrix Analysis) analysis to calculate protease activities [297], enabling simultaneous monitoring of multiple proteases. To further boost the performance and decrease the material requirements, a lab-on-a-chip assay was developed for simultaneous monitoring of multiple MMPs and ADAMs in a breast cancer cell line [298]. While integration of protease-responsiveness into lab-on-a-chip modalities is still limited, such multiplexed and highly parallel platforms could potentially expedite the diagnostic profiling of cancer-related proteases.

## 4. Leveraging Protease Activity in Drug Delivery for Cancer Therapy

One of the key challenges in cancer treatment with potent drugs is to ensure that their influence is focused on the tumor while minimizing off target effects. Drug delivery paradigms have been designed for selective activation of therapeutic agents based on physical and chemical cues from the tumor microenvironment that differ from elsewhere in the body, such as pH or redox potential [299]. Pathologic activity of proteases represents another biochemical feature of the tumor microenvironment that can be leveraged for this purpose. A growing number of protease-sensitive nanosystems and nanomaterials utilize disease-associated protease activity for their activation, selective targeting, or on-site release of drug payloads [300,301]. These strategies (Figure 2, right) serve to enhance specificity, improve targeting, decrease off-target effects, and increase the therapeutic index of a drug, thus enhancing its efficacy. In this section, therapeutic modalities that centrally integrate protease responsiveness are reviewed.

### 4.1. Prodrugs

The first group are the protease-activated prodrugs that use protease cleavage to release or activate the drug on-site. They integrate knowledge of disease-specific proteases to incorporate a specific cleavable sequence that ensures release dependent on the activity of a specific target protease. Leucine-doxorubicine prodrug [302] offers perhaps the simplest example to demonstrate the utility of the concept, and showed promising results in tumor models. In this case, a short peptide attached to the drug needs to be cleaved by the target protease for the drug to be activated. While this prodrug is rather unspecific, there are other examples that activate in response to more specific protease cleavage. These include the legumain-sensitive Ala-Ala-Asn linker [303], cathepsin B sensitive linker Val-Cit [304], and several other prodrugs requiring different MMPs, kallikreins, cathepsins, and coagulation proteases, among others [305].

### 4.2. Antibody Drugs

Antibody-drug conjugates constitute another group of therapeutics that rely on proteolysis in the tumor. In this case, the antibody recognizes a cancer cell-specific molecular feature (e.g., cell surface receptor) and is eventually internalized into the target cell [306,307]. The conjugate is subsequently processed by lysosomal proteases and released from the lysosome, destroying the target cells. There are currently two antibody-drug conjugates activated by cleavage of the Val-Cit linker on the market, ADCERTIS [308] and Polivy [309], both of which are approved for cancer treatment. Presently, there is considerable interest in leveraging this concept to develop cleavable linkers for other proteases to improve the serum stability and on-site release of drugs. While the use of antibodies is beneficial for achieving targeting and specificity, antibodies can also have unwanted effects and further development is needed to improve selectivity toward targeted cells [310]. Protease cleavage-activation can be also integrated into therapeutic antibodies to help overcome this challenge. This concept was recently incorporated in Ab prodrugs (Probodies^TM^) by CytomX Therapeutics [311]. Their probody targets the EGF receptor highly expressed on the surface of cancer cells, while incorporating a protease-cleavable peptide into cetuximab to boost selectivity. This peptide needs to be cleaved by legumain, uPA, or matriptase to activate the probody and has shown promising preclinical results [312]. Of note, application of Probodies^TM^ to preclinical imaging is generally simple and could be applied in multiplexed settings, thus potentially aiding the evaluation of different proteases as prospective cancer biomarkers.

### 4.3. Polymers

Responsiveness to disease-related proteases can also be incorporated into polymer architectures, including the biodegradable and biocompatible polymer-based nanoparticles that have been investigated over the last decade as drug delivery vehicles [313]. Targeting of polymeric nanoparticles to tumors has been thought to rely on the EPR (Enhanced permeability and retention) effect, exploiting the leaky vasculature and poor lymphatic drainage [314], however the practical relevance of this effect in determining the fate of injected nanomaterials has been increasingly called into question [315]. Only a tiny fraction of an injected dose of nanomaterial reaches tumors, with one recent metaanalysis suggesting a median delivery efficiency of 0.7% of the initial dose [316]. This underscores the importance of strategies for site-specific release or activation in tumors for improving selectivity.

Polymers were integrated into various therapeutics to date [317]. One of the most widely used polymers in nanoscale therapeutic architectures for cancer is PEG (Poly(ethylene glycol)) [318], favored mostly due to its solubility and biodegradability, though the occurrence of anti-PEG antibodies in patients is an emerging issue that may eventually cause PEG to be supplanted by similar alternatives [319]. For an example of a PEG-based protease-responsive nanotherapeutic, PEG-functionalized QDs clusters sensitive to MMP-2 were developed to enable multistage penetration into tumor tissue [320], with their initial 100 nm size promoting accumulation in the tumor, followed by cleavage into 10 nm particles that could more readily diffuse in the tumor. Similarly, utilizing cancer-associated MMP-2 activity, a PEG 2000-paclitaxel conjugate showed improved tumor targeting characteristics over standard paclitaxel [321]. Another interesting example includes SELPs (Silk-elastin-like protein polymers). Since several MMPs exhibit strong elastase activity, these polymers could serve as specific MMP-responsive delivery agents for tumor targeted drug delivery [322]. Also poly(L-glutamic acid) conjugated with paclitaxel enabled a cathepsin B-dependent drug release, showing promising results in preclinical ovarian cancer models [323]. Similar conjugates were developed for thrombin [324] and trypsin [325]. A further polymer successfully used for preparation of polymer-drug conjugates with protease-responsiveness is HPMA (N-(2-hydroxypropyl)methacrylamide copolymer) [326]. In this case, the conjugate incorporated a PSA-cleavable peptide to release thapsigargin, a natural cytotoxin, and the system showed promising results in a prostate cancer model. Like the polymer drug delivery vehicles, the serum protein albumin has been successfully used as a macromolecular carrier in the preparation of albumin-drug conjugates. Several conjugates with cancer therapeutics and cleavable protease linkers were developed for cathepsin B [327] and PSA [328] among others. These efforts corroborate protease-sensitive macromolecular drug conjugates as a viable strategy for improved on-site drug targeting.

### 4.4. Liposomes

Liposomes are yet another type of delivery system with oncologic applications. Their advantages include a size range suitable for accumulation in tumors and surface chemistry readily adaptable to various functionalities [329]. In a common approach, the drug (e.g., cytotoxic compound) is encapsulated in liposomes functionalized with protease-sensitive ligands. These protease-responsive liposomes have been integrated with cancer gene therapy to boost the efficiency of cancer cell transduction [330]. One such design relied on a PEGylated MMP-cleavable peptide serving as a steric hindrance to regulate liposome cell entry. Highly-targeted liposomes can be also designed by combining the advantages of multifunctional liposomes and protease-sensitive polymers to generate polymer-caged liposomes [331]. Here, a graft copolymer of poly(acrylic acid) containing a peptide with a cleavage site for the cancer protease uPA was used to crosslink liposomes. These liposomes showed excellent stability and efficient cargo release upon uPA cleavage, making them a promising cancer targeting modality. Recently, another version of liposomes using PEGylated MMP-cleavable lipopeptide was reportedly used to improve the cytotoxicity of anticancer drugs [332]. Liposomes can be also coated with cell-penetrating TAT (Transactivator of transcription) peptides that are modified with protease cleavage sequences, as demonstrated by a legumain-activated liposome for tumor targeting [333]. In this case, the tumor-expressed legumain cleavage ensures improved efficiency in the delivery of doxorubicin to tumors. A further strategy to generate protease-responsive liposomes is the destabilization of liposome integrity with ”uncorking” that is achieved by integrating MMP-9-sensitive triple helical lipopeptides that result in cargo release upon cleavage [334]. Additionally, a newly emerging concept in the liposome field is the use of proteases not directly as activators, but rather as targeting moieties. For example, a liposome drug delivery system was reported where the liposomes were coated with a selective cathepsin B inhibitor [335]. While cathepsin B resides in lysosomes under normal physiological conditions, it is highly expressed on the surface of cancer cells and can be used for tumor targeting.

### 4.5. Inorganic Nanomaterials

Finally, systems based on inorganic nanomaterials that incorporate responsiveness to proteases for therapeutic effects have also seen recent advancements. Frequently, this includes various silica and iron oxide nanoparticles that can be integrated with protease-responsive elements to improve cancer targeting. Such approaches have already found application in the field of protease therapeutics and have already been extensively reviewed elsewhere [244,336,337,338]. As a representative example, in one study, mesoporous silica nanoparticles were packed with doxorubicin and coated with gelatin to prevent drug leakage. After accumulating in tumors, elevated activity of MMP-2 broke down the gelatin layer, releasing the drug cargo [339]. Nanomaterials may offer additional advantages if they can be developed into theranostic agents, which simultaneously integrate dual modalities as diagnostic reporters and targeted drug delivery systems [244]. One such example is the use of iron oxide nanoparticles prepared by conjugating ferumoxytol to an MMP-activatable peptide linked with azademethylcolchicine [340]. This theranostic system efficiently induced apoptosis of cancer cells in a breast cancer model, and also enabled precise monitoring of its biodistribution via T2 contrast. Although current work tends to emphasize ultrasmall iron oxide nanocrystals that act as T1 contrast agents [341], this example is broadly illustrative of the role that protease responsiveness can play in integrating multiple modalities into a single theranostic agent.

## 5. Conclusions and Outlook

Exploration of the protease landscape associated with cancer has enabled the emergence of promising clinical technologies exploiting pathologic patterns of proteolytic activity for diagnosis and therapy (Table 2). Engineering toolsets for incorporating protease responsiveness are more readily available than ever, with the potential to be integrated into nanosystems that empower the next generation of protease-based diagnostic and therapeutic modalities [342]. Protease research has already greatly benefited from interdisciplinary connections of molecular biology and biomedicine with nanotechnology, polymer chemistry, and microfluidics. These efforts strive toward a shared ideal of technologies that are sensitive, robust, and able to measure protease activity in real time, yet simultaneously cost-efficient, easily multiplexed, and practical to use at the point-of-care. We are convinced that the field of oncology will advance alongside the development of the next generation of protease sensors, protease-responsive nanomaterials, and lab-on-a-chip applications. While several of these technologies have already shown promising preclinical results for cancer diagnosis, disease staging, therapy stratification, and evaluation of therapeutic response, future developments are poised to unleash the full potential of proteases for oncologic applications.

## Figures and Tables

**Figure 1 ijms-22-02514-f001:**
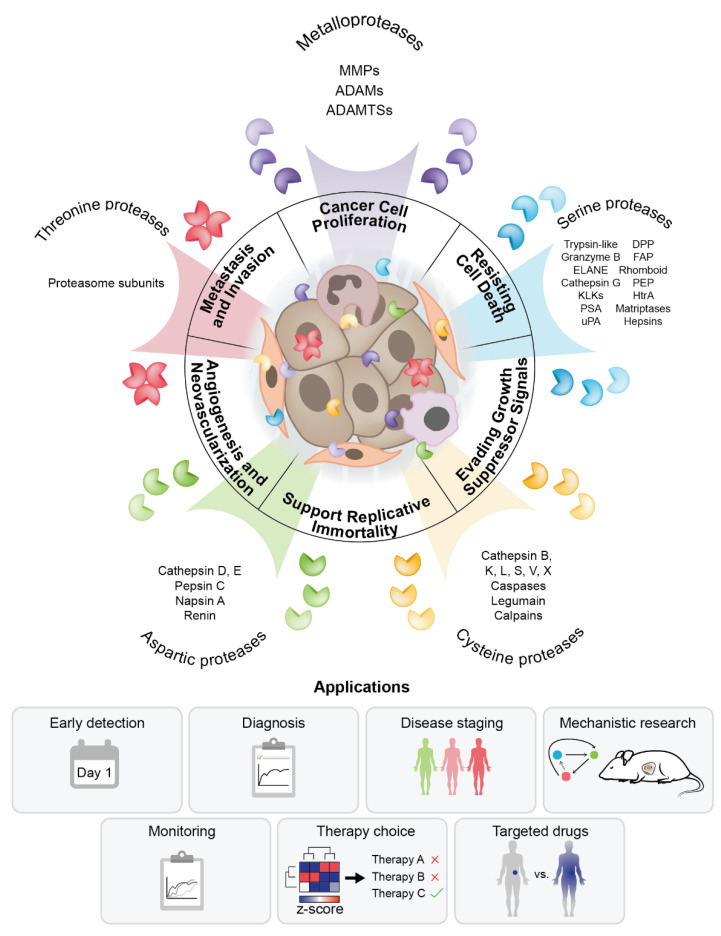
Knowledge of tumor proteolysis translates into diagnostic and therapeutic modalities. Proteases in the tumor microenvironment belong to different protease families and originate from tumor and other cells present in the tumor microenvironment. They exhibit different patterns of localization and activity that may overlap and generate an intricate proteolytic landscape with roles in mechanisms behind cancer hallmarks. Knowledge of the role of proteolysis in cancer can support diagnosis, staging, mechanistic studies, disease monitoring and therapy.

**Figure 2 ijms-22-02514-f002:**
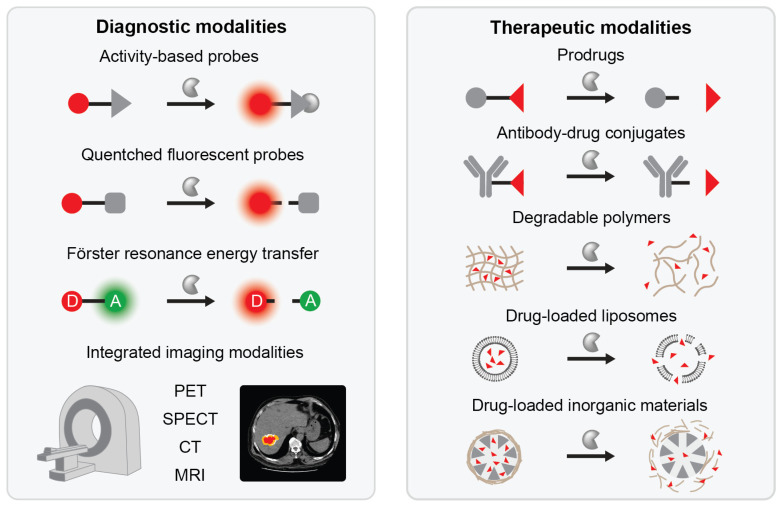
Protease-responsive nanodevices and applications of protease-responsiveness. Protease-responsiveness can be centrally integrated into various diagnostic modalities that utilize optical, MRI, PET, SPECT or CT imaging for detection of protease activity (left). Protease-responsiveness can also serve as a trigger for activation of prodrugs and antibody-drug conjugates or on-site targeting of cancer drugs via polymeric nanoparticles, liposomes and inorganic nanoparticles (right).

**Table 1 ijms-22-02514-t001:** Protease roles in the context of cancer hallmarks. Representative examples and selected roles of the most important proteases and protease groups are described with respect to cancer development and progression.

Cancer Hallmark	Example Proteases and Protease Groups	Mechanistic Roles, Functions and Consequences	Selected References
Cancer cell proliferation	MMP2, 3, ADAM10, 17 Cathepsins Kallikreins	ECM remodeling, signaling, processing of growth factors, sustain and boost proliferative signaling pathways	[33,34,35,36] [18,37] [38,39]
Resisting cell death	MMP7, ADAM10 Granzyme B HtrA Caspases Cathepsins B, L, S Cathepsin D	Apoptosis signaling, apoptosis resistance, circumvent apoptosis triggers, autophagy recycling, immune system evasion	[22,40] [41,42] [43] [44,45] [18,37] [46]
Evading growth suppressor signals	Various MMPs and ADAMs ADAMTs Kalikreins, Cathepsins	Cytokine and chemokine secretion, removal of cellular brakes, receptor signaling, disruption of p53 signaling	[22,36,40,47,48] [49,50,51] [39,52,53] [54,55]
Support replicative immortality	Multiple MMPs ADAM10, 17 Granyzme B Cathepsins B, L, S KLK4-7	Sustain growth signaling, release of biologically active fragments, immune system evasion, immune system hijacking	[33,56,57] [36,47,48] [41,42] [37,54,58,59] [60]
Angiogenesis and neovascularization	MMP1, 2, 9 Kallikreins PSA Cathepsins B, L, K, S Calpains	Growth factor signaling, ECM remodeling, degradation of structural proteins, release of cytokines, receptor shedding	[61,62,63] [52,53] [64] [37,54,59,65] [66,67]
Metastasis and invasion	MMP1, 14 ADAM10, 17 Kalikreins Cathepsin G PSA FAP, DPPIV, PEP Cathepsins B, L, K, S Legumain Cathepsin D, E Calpains	ECM remodeling, barrier degradation, cancer cell migration, receptor signaling, metabolic signaling, epithelial-to mesenchymal transition, release and modulation of signaling molecules, kinase signaling perturbation	[61,62,63] [36,47,48] [52,53] [68,69] [64] [70] [37,54,58,59] [71] [46,72] [73,74]

**Table 2 ijms-22-02514-t002:** Protease-responsive nanodevices for diagnostic and therapeutic purposes, application notes (describing advantages and shortcomings of the indicated modalities) and selected examples of use.

Protease-Responsive Nanodevices	Application Notes (+ Advantages, − Shortcomings)	Selected Examples and References
Activity-based probes	+ Sensitive detection of proteases in situ in cells and animal models + Excellent selectivity for target proteases − No signal amplification	Metalloproteases [213,214,215], serine proteases [216] including neutrophil proteases [217] and inflammation-related serine proteases [218], cysteine cathepsins [219,220], caspases [221,222] and legumain [223,224], aspartic proteases [225], and proteasome [226,227]
Protease-cleavable fluorescent substrate probes	+ Signal amplification + Selectivity can be improved by designs that incorporate unnatural amino acids − Background fluorescence and signal diffusion	Profiling of caspases [222], cathepsins [238,239], neutrophil proteases [217] and kallikreins [240]
QDs	+ Sensitivity of integrated FRET + Versatile platform − Toxicity of nanoparticles	Imaging and detection MMPs [247,248], caspase-1, collagenase, chymotrypsin and thrombin [249], uPAR [250], caspase-3 [251] and kallikrein [252]
PET and SPECT probes	+ High sensitivity and resolution − Short half-life of reagents because of radioactive isotopes − Costly detection modalities	Protease-responsive contrast agents for metalloproteases [263,264,265], cysteine cathepsins [266,267,268] and caspases [269,270]
MRI probes	+ High sensitivity and resolution for soft tissues − Expensive detection modality	MMP-2 in tumors [274], caspase activity after drug-induced apoptosis [275], caspase-3 [276], digestive elastases [277], cysteine cathepsins [279] and furin [280]
CT probes	+ Resolution for in-depth tissue imaging − Lack of suitable protease-sensitive probes	Protease-targeted iodinated probe [272] and protease activity probes with gold nanoparticles [273] were for cysteine cathepsins
Dual modality probes	+ Improved spatiotemporal resolution and sensitivity − Expensive and do not overcome the problems of original modalities	NIR/PET probes sensitive for MMP-2, MMP-9, and MMP-13 [281], FRET/SPECT probe for MMP-2 [282] and a FRET/MRI probe for MMP-2 [283]
DARPins	+ Selectivity for target protease + Can be integrated with other modalities − Intensive development and selection process	Imaging of cathepsin B in breast cancer [288]
Synthetic biomarkers	+ Sensitive in situ detection of protease activity + High multiplexing capabilities − Not best-suited for on-site monitoring (i.e., imaging)	Colorectal cancer biomarker detection in plasma [289], magnetically actuated protease sensors (MAPS) for measuring MMP tumor profiles in colorectal cancer [290], activity-based nanosensors (ABNs) for MMP-9 [291], Ultra small gold nanoclusters (AuNC) as MMP-responsive nanosensors [292], prostate cancer nanosensors [293] and lung cancer nanosensors [294]
Prodrugs with protease-cleavable linkers	+ Protease-dependent on-site activation − Off-site drug release due to unspecific protease cleavage	Legumain-sensitive Ala-Ala-Asn linker [303], cathepsin B-sensitive linker Val-Cit [304] and prodrugs that require metalloproteases, kallikreins, cathepsins or coagulation proteases [305]
Ab-drug conjugates	+ Improved on-site targeting of the drug − Problems with selective cancer cell recognition	ADCERTIS [308] and Polivy [309]
Probody	+ Protease-dependent on-site activation of the Ab − Non-selective peptide linker cleavage	EGFR targeting probody activated by legumain, uPA or matriptase cleavage from CytomX Therapeutics [311]
Polymers	+ Favorable biologic properties of polymers + Accumulation on target site due to EPR effect + Selective on-site release of polymer-bound drug by the target protease − Prolonged retention and off-site accumulation − Non-selective protease degradation of the polymers − Anti-polymer antibodies arise in patients	PEG-functionalized MMP-2-sensitive QDs clusters [320], MMP-2-sensitive PEG 2000-paclitaxel conjugate [321], SELPs (Silk-elastin-like protein polymers) polymers as MMP-responsive delivery agents [322], cathepsin B sensitive poly(L-glutamic acid)-paclitaxel conjugate [323] as well thrombin-sensitive [324] and trypsin-sensitive [325] conjugates, a PSA-sensitive HPMA-thapsigargin conjugate [326]
Liposomes	+ Small size and favorable biological properties + Can be integrated with various other modalities − Accelerated clearance from cardiovascular system − Potential for allergic reactions	PEGylated MMP-sensitive lyposome [330], polymer-caged uPA-sensitive liposomes [331], PEGylated liposome with a MMP-cleavable lipopeptide [332], TAT peptide bearing legumain-activated liposome [333], a MMP-9-sensitive ‘uncorking’ liposome [334], a cathepsin B targeting liposome [335]
Inorganic materials	+ Biocompatible materials with various shapes and sizes + Can be integrated into theranostic agents − Nanoparticle toxicity or off-site accumulation	MMP-responsive silica [339] and iron oxide nanoparticles [340] for targeted drug release

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
