# Peer review of "The Tumor Proteolytic Landscape: A Challenging Frontier in Cancer Diagnosis and Therapy"

_ijms, 2021, doi:10.3390/ijms22052514_

Round 1

Reviewer 1 Report

The review manuscript entitled “The tumor proteolytic landscape: A challenging frontier in cancer diagnosis and therapy” which is contributed by Vizovisek et al. describes an elegant manuscript which briefly discuss protease role in cancer and their potential role in diagnosis and therapy. However, some information need to update or enhance that will make this review manuscript has more impact in cancer translation research fields.

Major comments:

  1. The cancer hallmarks are revised in the 2011. Now, there are 10 parameters such as sustaining evading growth suppressors, proliferation signaling, deregulating cellular energetics, resisting cell death, genome instability & mutation, inducing angiogenesis, activating invasion & metastasis, tumor-promoting inflammation, enabling replicative immortality, avoiding immune destruction where the cellular protease should contribute in those 10 field. The author should integrate them into figure 1 and table 1.
  2. Since protease is cleavage substrate to generate new protein or peptide or degrade protein. In table 1, the authors should list the important target genes which is cleavage by individual proteases in the critical cancer hallmarks.
  3. Notch signaling pathway and sonic hedgehog are critical in tumorigenesis and progression. The protease activity is keystone in both signaling activation. The authors should discuss them in the manuscript.
  4. The authors should summary the substrate domain or conserve sequence of individual protease which can help reader to apply in their activity and substrate-based studies and also increase the value of review manuscript.

Author Response

The review manuscript entitled “The tumor proteolytic landscape: A challenging frontier in cancer diagnosis and therapy” which is contributed by Vizovisek et al. describes an elegant manuscript which briefly discuss protease role in cancer and their potential role in diagnosis and therapy. However, some information need to update or enhance that will make this review manuscript has more impact in cancer translation research fields.

Major comments:

  1. The cancer hallmarks are revised in the 2011. Now, there are 10 parameters such as sustaining evading growth suppressors, proliferation signaling, deregulating cellular energetics, resisting cell death, genome instability & mutation, inducing angiogenesis, activating invasion & metastasis, tumor-promoting inflammation, enabling replicative immortality, avoiding immune destruction where the cellular protease should contribute in those 10 field. The author should integrate them into figure 1 and table 1.

REPLY: We thank the reviewer for his or her comment, however we respectfully disagree. While the paper by Hanahan and Weinberg from 2011 (DOI: 10.1016/j.cell.2011.02.013) indeed provides an update to the cancer hallmarks, not all hallmarks are equally relevant to proteases. Specifically, hallmarks like genome instability and mutation, as well as deregulated cellular energetics, are not directly connected to proteases. It is therefore difficult to justify adding these as separate categories to be discussed within the context of the roles of individual members of protease families. The reader is better served by focusing on the original six hallmarks of cancer and the directly associated roles of proteases, which are extensively discussed in the paper. We certainly acknowledge the additional hallmarks and specify that we are discussing the roles of proteases linked to selected hallmarks of cancer.

The modified paragraph now reads as follows:

"Representative members from each major protease family have been linked to selected hallmarks of cancer (Figure 1) and elevated or dysregulated protease activity is mechanistically involved in sustaining cancer cell proliferation, resisting cell death, evading growth suppressor signals, supporting replicative immortality, inducing angiogenesis and neovascularization and metastasis development and invasion [21, 30]. Accordingly, representative protease groups and their roles in context of the original six cancer hallmarks proposed by Hanahan and Weinberg are summarized in Table 1."

  1. Since protease is cleavage substrate to generate new protein or peptide or degrade protein. In table 1, the authors should list the important target genes which is cleavage by individual proteases in the critical cancer hallmarks.

REPLY: While we appreciate this comment, we would like to first clarify a few key concepts. Proteases usually cleave their substrate proteins as a part of their maturation process, specific activation/inactivation, functional/structural changes or degradation as a part of the lifecycle of a protein. While proteases generally do not interact with genes in cancer directly, there are different gene-regulated pathways that lead to upregulated/changed expression levels of proteases in cancer. Perhaps one of the best known such pathways is the NF-kB signaling, which is known to substantially contribute to the elevated protease pool in cancer. This is already described in the paper to a sufficient level of detail given its diagnostic and therapeutic protease focus. Moreover, proteases and genes also do not interact in a way that would directly result in cleavages. Nevertheless, it is important to note that proteases are indeed crucial for maturation of many proteins (i.e. product of their respective genes). Several examples are included in the paper, e.g. cytokine and chemokine maturation, processing of interleukins and activation of pro-forms of proteases, among others. We are therefore convinced that protease regulation is sufficiently covered and further expanding on this would go beyond the scope of the paper and imbalance its current focus.

  1. Notch signaling pathway and sonic hedgehog are critical in tumorigenesis and progression. The protease activity is keystone in both signaling activation. The authors should discuss them in the manuscript.

REPLY: We thank the reviewer for this insightful remark. The Notch pathway is indeed important for cancer development and progression, and we have added a short paragraph describing its links with prominent cancer proteases at the end of the ADAM section, since these are most often associated with Notch signaling. Regarding sonic hedgehog signaling, some clarification is needed. Hedgehog proteins are related to Zn2+-containing metalloproteases, but act as receptor ligands rather than as proteases (as described extensively by Maun et al., 2010, DOI: 10.1515/BC.2010.098). Considering their mechanism of action, hedgehog proteins contain pseudo-active sites that are essential for their binding to their respective receptors and have been therefore considered as a potential new target group for cancer therapy. Nevertheless, these roles are non-proteolytic and do not fit the scope of the paper.

For Notch signaling, we added the following sentences to the ADAM section:

"Different ADAMs, including ADAM-10 and ADAM-17, are also important for proteolytic processing of ligands and modulation of Notch signaling. This molecular mechanism requires ADAM-mediated cleavage events for its proper functioning and altered processing of Notch receptors and ligands has been linked to proliferation and differentiation as well as cancer cell death [72-74]."

  1. The authors should summary the substrate domain or conserve sequence of individual protease which can help reader to apply in their activity and substrate-based studies and also increase the value of review manuscript.

REPLY: Our review paper describes more than 100 proteases with links to cancer development and progression belonging to major families of metallo, serine, cysteine and aspartic proteases, among others. Each of these families has its own specific structural features that form the basis on which the family members are divided into different classes. These protease classes vary substantially by catalytic mechanism, individual domain architecture, structural discriminating features as well as activity, stability, physiological roles and substrate cleavage patterns. These are reviewed extensively in several recent publications, which are also cited in our review paper for comprehensiveness (see for example Alaseem et al., 2019, DOI: 10.1016/j.semcancer.2017.11.008, Cal et al., 2015, DOI: 10.1016/j.matbio.2015.01.013, Unruh et al., 2020, DOI: 10.1186/s13045-020-00932-z, or Vidak et al., 2019, DOI: 10.3390/cells8030264). The genomic and structural studies for proteases are usually done on a family-specific or class-specific basis, because comparing proteases of different classes beyond catalytic mechanisms is usually not very informative. The same is also true for protease substrate ensembles, which are the domain of more specialized proteomic, chemical biology or animal model studies. These cannot be generalized across different protease families and are usually performed on a protease class or a contextual group of a protease sub-family. While several protease substrate studies (e.g. degradomic substrate screens) that are relevant for the paper are readily cited, we believe that further expansion on this topic would divert from the focus of the paper. Furthermore, the diversity of the proteases covered by our paper is far too large to permit a visualization that would highlight their individual architectures in a meaningful way. Therefore, this topic is usually addressed in more specific review papers which we comprehensively cited to guide the reader with a specific interest in a particular protease class to the relevant publications (e.g. Kessenbrock et al., 2010, DOI: 10.1016/j.cell.2010.03.015, Turk et al., 2012, DOI: 10.1016/j.bbapap.2011.10.002). 

Reviewer 2 Report

In the present review, Vizovisek et al. provide a profound and detailed overview of the role of proteases in cancer development, diagnosis, and treatment. This work is of high and translational interest for basic biological research, diagnosticians, and treating physicians. It is based on profound literature research and compilation of the most recent and relevant works and scientific knowledge, supported by well-illustrated graphics and clear and comprehensive tables. Therefore, I endorse the publication of the review by Vizovisek et al. without any changes/revisions.

Author Response

In the present review, Vizovisek et al. provide a profound and detailed overview of the role of proteases in cancer development, diagnosis, and treatment. This work is of high and translational interest for basic biological research, diagnosticians, and treating physicians. It is based on profound literature research and compilation of the most recent and relevant works and scientific knowledge, supported by well-illustrated graphics and clear and comprehensive tables. Therefore, I endorse the publication of the review by Vizovisek et al. without any changes/revisions.

REPLY: We thank the reviewer for his or her assessment of our work. We look forward to sharing our review paper with the research community and hope that it will appeal to a broad readership.